# Mutational Spectrum and Clinical Features of Patients with *LOXHD1* Variants Identified in an 8074 Hearing Loss Patient Cohort

**DOI:** 10.3390/genes10100735

**Published:** 2019-09-23

**Authors:** Karuna Maekawa, Shin-ya Nishio, Satoko Abe, Shin-ichi Goto, Yohei Honkura, Satoshi Iwasaki, Yukihiko Kanda, Yumiko Kobayashi, Shin-ichiro Oka, Mayuri Okami, Chie Oshikawa, Naoko Sakuma, Hajime Sano, Masayuki Shirakura, Natsumi Uehara, Shin-ichi Usami

**Affiliations:** 1Department of Otorhinolaryngology, Shinshu University School of Medicine, 3-1-1 Asahi, Matsumoto, Nagano 390-8621, Japan; mkarunakyuudou@icloud.com (K.M.); nishio@shinshu-u.ac.jp (S.-y.N.); 2Department of Hearing Implant Sciences, Shinshu University School of Medicine, 3-1-1 Asahi, Matsumoto, Nagano 390-8621, Japan; 3Department of Otorhinolaryngology, Toranomon Hosipital, 2-2-2 Toranomon, Minato-ku, Tokyo 105-8470, Japan; abe3387@ybb.ne.jp; 4Department of Otorhinolaryngology, Hirosaki University School of Medicine, 53 Honcho, Hirosaki, Aomori 036-8203, Japan; goto-s@hirosaki-u.ac.jp; 5Department of Otolaryngology-Head and Neck Surgery, Tohoku University School of Medicine, 1-1 Seiryomachi, Aoba-ku, Sendai, Miyagi 980-0872, Japan; y-honkura@ktb.biglobe.ne.jp (Y.H.); msyukis.shrkrr.1987@gmail.com (M.S.); 6Department of Otorhinolaryngology, International University of Health and Welfare, Mita Hospital, 1-4-3 Mita, Minato-ku, Tokyo 108-8329, Japan; iwasakis@shinshu-u.ac.jp (S.I.); okashin@shinshu-u.ac.jp (S.-i.O.); 7Kanda ENT Clinic, Nagasaki Bell Hearing Center, 4-25 Wakakusa-cho, Nagasaki, Nagasaki 582-8023, Japan; n-bell@estate.ocn.ne.jp; 8Department of Otolaryngology-Head and Neck Surgery, Iwate Medical University, 19-1 Uchimaru, Morioka, Iwate 020-8505, Japan; ymkobaya@iwate-med.ac.jp; 9Department of Otorhinolaryngology, Tokai University School of Medicine, 143 Shimokasuya, Isehara, Kanagawa 259-1193, Japan; mayuri.okami@gmail.com; 10Department of Otorhinolaryngology-Head and Neck Surgery, Graduate School of Medical Sciences, Kyushu University, 3-1-1, Maidashi, Higashi-ku, Fukuoka, Fukuoka 812-8582, Japan; chieoshika@yahoo.co.jp; 11Department of Otorhinolaryngology, Yokohama City University Medical Center, 4-57 Urafune, Minami-ku, Yokohama, Kanagawa 232-0024, Japan; n_sakuma@yokohama-cu.ac.jp; 12Department of Rehabilitation, Kitasato University School of Alliked Health Sciences, 1-15-1 Kitasato Minami-ku, Sagamihara, Kanagawa 252-0375, Japan; sanohj@med.kitasato-u.ac.jp; 13Department of Otolaryngology-Head and Neck Surgery, Kobe University School of Medicine, 7-5-2 Kusunoki-cho, Chuo-ku, Kobe, Hyogo 650-0017, Japan; natsuko0522@gmail.com

**Keywords:** *LOXHD1*, non-syndromic hearing loss, DFNB77, recurrent variation, haplotype analysis, cochlear implantation

## Abstract

Variants of the *LOXHD1* gene, which are expressed in hair cells of the cochlea and vestibule, have been reported to cause a progressive form of autosomal recessive non-syndromic hereditary hearing loss, DFNB77. In this study, genetic screening was conducted on 8074 Japanese hearing loss patients utilizing massively parallel DNA sequencing to identify individuals with *LOXHD1* variants and to assess their phenotypes. A total of 28 affected individuals and 21 *LOXHD1* variants were identified, among which 13 were novel variants. A recurrent variant c.4212 + 1G > A, only reported in Japanese patients, was detected in 18 individuals. Haplotype analysis implied that this variation occurred in a mutational hot spot, and that multiple ancestors of Japanese population had this variation. Patients with *LOXHD1* variations mostly showed early onset hearing loss and presented different progression rates. We speculated that the varying severities and progression rates of hearing loss are the result of environmental and/or other genetic factors. No accompanying symptoms, including vestibular dysfunction, with hearing loss were detected in this study. Few studies have reported the clinical features of *LOXHD1*-gene associated hearing loss, and this study is by far the largest study focused on the evaluation of this gene.

## 1. Introduction

One of the most common sensory impairments is hearing loss (HL), affecting one in 500 to 700 newborns [1]. To date, roughly a hundred genes have been reported as associated with inherited non-syndromic HL, and at least half of prelingual HL cases account for genetic factors [1,2]. Among these reported genes, around 65% are a cause for autosomal recessive sensorineural hearing loss (ARSNHL), and over 70 causative genes have been identified [2]. The major causative genes of ARSNHL patients in Japan are *GJB2* (14.8%), *SLC26A4* (4.6%) and *CDH23* (3.4%) [3]. Less common causative genes such as *LOXHD1* are rarely detected in SNHL patients, making it difficult to diagnose. The detection of the cause of a patient with HL is relevant, since their etiology could enable us to predict their treatment outcome, and thus provide useful information to better assistant the decision making regarding their course of treatment and/or HL intervention. By introducing massively parallel DNA sequencing (MPS) screening of 68 deafness-causative genes, detection of novel variants of rare causative genes have simplified significantly.

*LOXHD1* is a known cause of DFNB77, a progressive form of ARSNHL. It is mapped on chromosome 18q12–21, and it encodes the protein Lipoxygenase Homology Domains 1, which consists entirely of 15 PLAT (polycystin-1, lipoxygenease, alpha-toxin) domains [4]. Grillet et al. previously reported in murine studies that Loxhd1 proteins are localized in hair cells along the plasma membrane of stereocilia, which helps maintain normal hair cell function [5]. They reported HL in *samba* mice due to the rapid degeneration of cochlear hair cells shortly after birth.

Studies have shown that the variations of the *LOXHD1* gene show progression of HL, leading to profound to severe non-syndromic HL. However, not many studies have reported its accompanying phenotypes, including the severity, progression and vestibular function, thus the correlation of genotype and phenotype is unclear. In this study, we detected a large group of HL patients with *LOXHD1* variations. Here, we report the novel variations detected by MPS and assess the phenotypes of patients with *LOXHD1* variation.

## 2. Materials and Methods

### 2.1. Subjects

For this study, a total of 8074 Japanese HL patients (autosomal dominant sensorineural hearing loss, 1336; ARSNHL, 5564; and inheritance unknown, 1174) were recruited from 67 otolaryngology departments nationwide described in our previous reports [6]. Among these subjects, we selected patients with *LOXHD1* variants by using MPS on 68 target genes (Appendix A). To participate in this study, a written informed consent was obtained from all patients (or from their next of kin, caretaker or guardian in case of minors or children), which was approved by the Shinshu University Ethical Committee and the ethical committee within each participating institution. Clinical information and peripheral blood samples were obtained from each subject and from all their consenting relatives.

This study was conducted in accordance with the Declaration of Helsinki, and the protocol was approved by the Ethics Committee of Shinshu University School of Medicine No. 387—4 September 2012, and No. 576—2 May 2017.

### 2.2. Variant Analysis

Amplicon libraries of the target exons were prepared with an Ion AmpliSeq Custom Panel (ThermoFisher Scientific, Waltham, MA, USA) designed using Ion AmpliSeq Designer for 68 genes reported to cause nonsyndromic hearing loss with the Ion AmpliSeq Library Kit 2.0 (ThermoFisher Scientific) and Ion Xpress Barcode Adapter 1-96 Kit (ThermoFisher Scientific) according to the manufacturer’s instructions [7,8]. The detailed protocol has been described elsewhere [8]. After the amplicon libraries were prepared, equal amounts of the libraries for six patients were pooled for one Ion PGM sequence reaction and those for 45 patients were pooled for one Ion Proton sequencing. The emulsion PCR was performed with the Ion OneTouch System and Ion OneTouch 200 Template Kit v2 (ThermoFisher Scientific) or Ion PI Hi-Q OT2 200 Kit according to the manufacturer′s instructions. Sequencing was performed with an Ion torrent PGM system using the Ion PGM 200 Sequencing Kit and Ion 318 Chip (Thermo- Fisher Scientific), or Ion Proton system using the Ion PI Hi-Q Sequencing 200 Kit and Ion PI Chip (ThermoFisher Scientific) according to the manufacturer′s instructions.

The sequence data were mapped against the human genome sequence (build GRCh37/ hg19) with a Torrent Mapping Alignment Program. After sequence mapping, the DNA variant regions were piled up with Torrent Variant Caller plug-in software. After variant detection, their effects were analyzed using ANNOVAR software [9]. The missense, nonsense, insertion/deletion and splicing variants were selected from among the identified variants. Variants were further selected as less than 1% of the 1000-genome database, the 6500 exome variants, the Human Genetic Variation Database (dataset for 1208 Japanese exome variants) and the 333 in-house Japanese normal hearing controls. Direct sanger sequencing was utilized to confirm the selected variants. Copy number analysis based on the read depth data obtained from NGS analysis was also performed for all 68 genes in accordance to our previous paper [8].

The allele frequency of a variant was evaluated by the Exome Aggregation Consortium (ExAC03) and the Genome Aggregation Database (gnomAD). The pathogenicity of a variant was evaluated by ACMG (American College of Medical Genetics) standards and guidelines [10]. For missense variants in particular, functional prediction software, including Sorting intolerant from Tolerant (SIFT), Polymorphism Phenotyping (PolyPhen2), Likelihood Ratio Test (LRT), Mutation Taster, Mutation Assessor and Combined Annotation Dependent Depletion (CADD) were used in the ANNOVAR software.

### 2.3. Clinical Evaluations

We collected the onset age and progressiveness of HL, pedigree, complaints of tinnitus and vertigo. Further vestibular examinations, e.g., measurement of cervical and ocular vestibular evoked myogenic potential (cVEMP and oVEMP) and caloric testing, were obtained from some patients. Evaluation of HL by pure-tone audiometry was performed on patients over 5 years old, and the auditory steady state response (ASSR), conditioned orientation response audiometry (COR; one of the behavioral audiometries) or play audiometry was performed on patients under 4 years old. The pure tone average (PTA) was calculated from the audiometric thresholds at four frequencies (0.5, 1, 2 and 4 kHz). We classified their HL into 4 categories: mild (PTA 20–40 dB), moderate (41–70 dB), severe (71–90 dB) and profound (>91 dB). The audiometric configurations were categorized to flat, low-frequency HL, mid-frequency HL, sloping high-frequency HL (gradually falling 10 dB for high frequency) and precipitous high-frequency HL (higher frequency thresholds worsen by at least 20 dB per octave) as reported previously [11].

Intervention outcome for HL, including the use of cochlear implants, was analyzed based on medical charts.

### 2.4. Haplotype Analysis

Haplotype pattern within the 2 Mbp region surrounding position hg19:chr18:44114297, for the frequent Japanese variation *LOXHD1*:NM_144612:c.4212+1G>A identified in this study, was analyzed using a set of 32 single nucleotide polymorphisms (SNPs) (16 sites upstream and 16 sites downstream). For this analysis, we selected 6 patients with this homozygous variation. Haplotype analysis was performed by the direct sanger sequencing.

## 3. Results

### 3.1. Detected Variations

We identified 21 possibly disease-causing *LOXHD1* variants, 13 of which were novel variants (Table 1). All individuals with novel variants belonged to independent pedigrees except for 1 sibling (Figure 1). The novel variants consisted of 7 missense variants, 1 nonsense variant, 4 splicing variants and 1 frameshift deletion variant. The allele frequency of all variants was less than 0.01% using the aforementioned database. Based on the ACMG guidelines, the novel variants were categorized into 2 pathogenic, 4 likely pathogenic and 7 uncertain significance. The segregation analysis is not complete on some pedigrees since we could not obtain peripheral blood samples from all the family members. This could indicate that the *LOXHD1* variants could possibly not be the cause of the patient′s HL, yet, our filtering process detected no other biallelic candidate variants of other recessive genes from the MPS results. Specifically, families #21 and #22 looks to be possibly of dominant or X-linked heredity, and only the *LOXHD1* mutations remained after filtering, thus we included these two cases. To further confirm that two *LOXHD1* variants are in *trans* allele and be the causative gene of their HL, further analysis by the long-read sequencing could be considered. Notably, among the eight previously reported *LOXHD1* variants that were also identified in this study, the c.1828G > A and c.3281A > G variants had relatively high carrier frequencies in the Japanese control population database (Human Genetic Variation Database (HGVD) and 3.5KJPN-Integrative Japanese Genome Variation Database) [12,13]. The minor allele frequency (MAF) of c.1828G > A in the Japanese control was 0.7% (70/9510 allele), whereas c.3281.A > G was 0.2% (2/9454 allele). These carrier frequencies were relatively higher compared to those of other population datasets (ExAc, gnomAD). Thus, the pathogenicity of these two variants are controversial.

### 3.2. Clinical Features of Patients with LOXHD1 Variants

In this study, 28 affected individuals from 25 families were identified, which include 3 previously reported individuals [15,21]. Their clinical findings are summarized on Table 2. 

The onset age of their HL varied from 0 to 36 years old (mean age: 6.1 years), among which 14 patients had congenital HL. Only four individuals noticed their HL post childhood. A total of 6 out of 21 patients (28.5%) complained of tinnitus, and none of vertigo. A total of 15 out of 28 individuals (53.6%) were aware of progression of HL at the time of their genetic testing. To further assess the progression of HL, we evaluated the deterioration of hearing by obtaining serial audiograms from 14 individuals (Figure 2). The deterioration rate varied from 0.1 dB/year to 9.3 dB/year in PTA (mean rate: 3.5 dB/year). This further follow up revealed progression of HL in four individuals unaware of deterioration of HL at the time of their genetic testing.

We acquired audiograms from 27 individuals (Figure 1). All individuals presented bilateral HL, and 25 presented symmetrical HL, except for two individuals (family no. 13 and 21). The audiometric configurations were classifiable into 14 flat types, 8 sloping high-frequency HL and 6 precipitous high frequency HL audiograms. Their severity of HL at the point of genetic examination varied from mild to profound, including 3 mild HL (11%), 7 moderate HL (26%), 6 severe HL (22%) and 11 profound HL (41%).

### 3.3. Recurrent Variant

A recurrent variation, c.4212 + 1G > A, was identified in 9 individuals with this homozygous variant and in 9 with this compound heterozygous variant. Between these variants, there was no specific difference in the phenotypes. When compared age wise, older individuals showed a significant deterioration of HL (Figure 3). Specifically, patients in their youth showed varying degrees of HL from mild to profound HL, whereas patients over adolescence all showed severe to profound HL.

This variant has only been reported in the Japanese HL patients (Table 3). To evaluate whether this variant is a common ancestor phenomenon or mutational hot spot, we performed a haplotype analysis on six homozygous variant patients, which presented differing haplotypes (Table 4). 

Though less in number, c.3281A > G was another recurrently identified variation. Four individuals with this compound heterozygous variation were identified in this study. Two individuals (family no. 16 and 19) with missense compound heterozygous variations showed late onset of over 30 years old (Table 2). The other two individuals (family no. 11 and 23) had compound heterozygous variations with the other recurrent c.4212 + 1G > A, and they both presented early onset severe HL. All individuals with this variation exhibited severe to profound HL at the time of their genetic testing (Figure 1, family no. 11, 16, 19 and 23). This variation has previously been reported among the Chinese population, yet no clinical characteristics were presented in the paper [16].

### 3.4. Intervention

Of the 28 individuals, eight individuals were reported to have received cochlear implantation (CI). We followed up on four of these individuals (family no. 1, 6, 13 and 22) after CI. We were able to obtain complete results of the Japanese monosyllable, word and sentence perception tests pre-CI from two of these four patients. These two recieved CI in their adulthood, and showed a precipitous type of audiogram configuration. All CI patients showed a favorable outcome and showed over 90% correct six months post-operation (Figure 4). 

## 4. Discussion

In this study, we identified 28 individuals with *LOXHD1* variants including 13 novel variants by MPS screening. A total of 44 *LOXHD1* variants have been reported to date from various countries, and this study is the largest analysis of *LOXHD1*-related HL yet (Table 3) [5,14,15,16,17,18,19,20,21,22,23,24,25,26,27,28,29,30,31,32,33,34]. Among the identified variants, 11 were non-truncating variations (52.4%) and 10 were truncating variations (47.6%). This ratio is consistent with the previously reported 26 non-truncating variations and 18 truncating variations (Table 3).

The prevalence of *LOXHD1*-gene associated HL was 0.365% (28/8074) of SNHL probands and 0.50% (28/5564) of ARSNHL probands in this Japanese population. The estimated frequency of Japanese HL patients with *LOXHD1* c.4212 + 1G > A homozygous variation, calculated from the carrier frequencies observed in HGVD and 3.5KJPN [12,13], was 0.14%. Our studies showed the frequency of this homozygous variation to be 0.1% (8/8,074), which support the accuracy of the prevalence estimated from this study [34,35]. The prevalence of this gene is notably lower than that of other more common ARSNHL causing genes, such as *GJB2*, *SLC26A4* and *CDH23*. Few previous reports have discussed the prevalence of the *LOXHD1* gene amongst SNHL patients. In relatively large cohorts that evaluated more than 100 patients, the prevalence of this gene was estimated to 0.71% in American SNHL patients (8/1119), 1.5% in the Netherlands (3/200) and 0.97% in Italy (1/103) [19,22,28]. This result is similar with our study in the American patients, but noticeably higher in the other two countries. Yet, there are limitations to discuss these differences because of the variance of their cohort sizes.

We analyzed the phenotypes of *LOXHD1* gene variation from various points. In this study, 85.7% (24/28) had an early onset HL. Among the four individuals with late onset HL, two had the same variation, c.3281A > G, and they were the only two cases with two missense compound heterozygous variations. The individuals who did not have this variation (family no. 21 and 22) both presented precipitous high-frequency HL at the time of their genetic testing. This suggests that their deterioration in hearing might have been undetected due to the lack of neonatal hearing screening or lack of regular audiometric examination. Previous reports mainly support that HL due to *LOXHD1* is congenital or of early onset, but some of the patients also showed late onset HL [20,27].

We identified that 64.3% (18/28) patients presented progression of their HL, and serial audiograms of several individuals revealed the deterioration rate of HL varying from slow to rapid declining HL. Interestingly, we identified that individuals with c.4212 + 1G > A, a recurrent variation only reported in Japanese HL patients (Table 4), show relatively rapid progression in HL and most of their HL deteriorated to severe to profound HL in their first decade (Figure 3). There have been controversial arguments to whether *LOXHD1* variations present progression in HL [5,14,15,16,17,18,19,20,21,22,23,24,25,26,27,28,29,30,31,32,33,34]. This study has revealed that most cases showed progressive HL or congenital deaf and some showed relatively stable HL. The HL progression of *LOXHD1*-associated HL may individually differ.

We also compared the severity of HL by type of mutation (non-truncating vs. truncating), by classifying the variations into PLAT domains, and by age. The results were inconclusive, and even in the patients with the same variations showed various phenotypes, especially, regardless of the same pedigree (family no. 8). We conclude that the varying phenotypes may be due to other environmental or genetic factors.

No individuals exhibited accompanying symptoms to *LOXHD1* related HL in this study. Riazzudin et al. have reported this gene mutation to be a cause of late onset Fuchs corneal dystrophy [35], yet there were no cases that distributed optic symptoms in this study. Grillet et al. previously reported, though not as prominent as in the cochlear stereocilia, *Loxhd1* is expressed in the stereocilia of vestibular hair cells, though no vestibular symptoms were observed in the murine study [5]. They observed *Loxhd1* expression to become prominent when cochlear stereocilia shed the kinocilium in its development, and suggested that the Loxhd1 protein help stabilize the stereociliary bundle in the absence of the kinocilium. They hypothesized that the discrepancy between *Loxhd1* expression in vestibular stereocilia and lack of vestibular function is because the vestibular stereocilia do not shed its kinocilium. In this study, no patients complained of vertigo, and also in further vestibular examinations, cVEMP, oVEMP and caloric testing of six individuals (family no. 1, 6, 13, 22, 23) showed no vestibular dysfunction (Table 2). Wesdorp et al. have also published results similar to this study that among the 14 individuals with *LOXHD1* variants that they identified, none revealed abnormalities under ophthalmological and vestibular evaluation [14]. These clinical findings might support Grillet’s hypothesis.

A recurrent variant c.4212 + 1G > A was detected in 64.3% (18/28) individuals in this study. This variant has only been reported among the Japanese HL patients. This insinuates that this variation is a mutation hot spot, or a common ancestor phenomenon. Our haplotype analysis revealed differing haplotypes, suggesting that it is a mutation hot spot, or that multiple ancestors of Japanese population had this mutation. On the gnomAD database, this variation has been detected in Africans, East Asians and non-Finnish Europeans, which could also support this hypothesis. No other variations have previously been reported to recur by the common ancestor phenomenon.

Eppsteiner et al. hypothesized that HL patients due to gene variations that are expressed in the membranous labyrinth produce better results with CI than patients that have HL due to gene variations expressed in the spiral ganglion [20]. As they also reported, since *LOXHD1* are expressed in cochlear stereocilias, it can be predicted that patients with this gene variation are good candidates for CI. This study showed that all patients with the *LOXHD1* variation who received CI showed favorable outcomes; all four cases for which we were able to obtain the results of the Japanese monosyllable word and sentence perception tests, showed over 90% correctness.

In conclusion, by analyzing the largest number of patients with *LOXHD1* related HL yet to be reported, we determined several characteristics of *LOXHD1* variations, and recurrent variants. In most cases, early onset is to be expected in patients with this gene variant. Rapid progression is commonly observed in patients with the c.4212+1G>A variation, yet other variants show a different rate of deterioration in HL. In this study, no symptoms accompanied HL with *LOXHD1* variations. The varying phenotypes can be the result of environmental and/or other genetic factors. All patients who received CI showed favorable outcomes regardless of their pre-operative clinical features, and it can be predicted that patients with *LOXHD1* variations are good candidates for CI.

## Figures and Tables

**Figure 1 genes-10-00735-f001:**
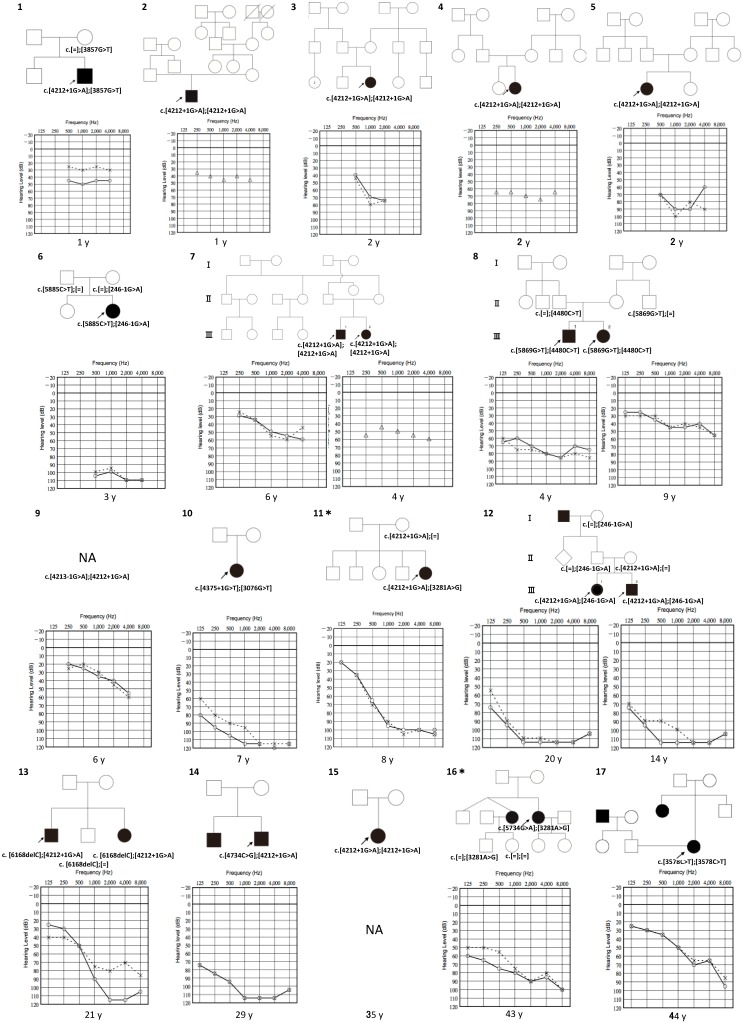
Pedigree and audiograms for each family with *LOXHD1* variants. Arrows show the probands in each family (family numbers 1–25). Genetic findings for each individual tested are noted in the pedigree. The age at the time of their genetic and audiometric testing is noted below the audiogram. ＊ These families include the variations mentioned in the footnote of Table 1.

**Figure 2 genes-10-00735-f002:**
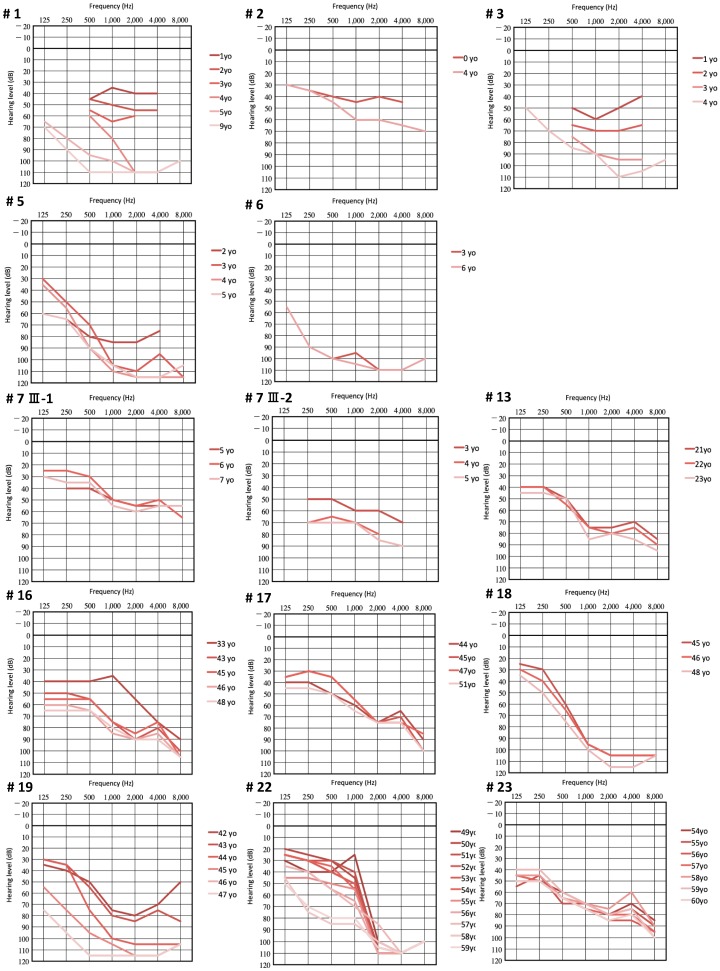
Serial audiograms of eight individuals with *LOXHD1* variations. Darker colors indicate the hearing thresholds in younger ages, and lighter colors indicate those in older ages.

**Figure 3 genes-10-00735-f003:**
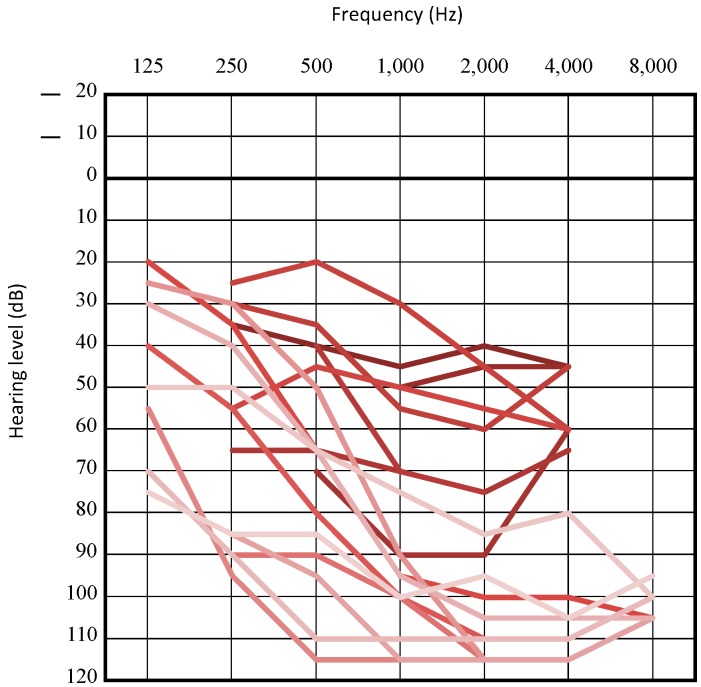
Overlapping audiograms of individuals with c.4212 + 1G > A variation. Pure-tone audiograms or conditioned orientation response audiograms were utilized. Darker colors indicate the hearing thresholds in younger ages, and lighter colors indicate those in older ages. Their ages ranged from 0 to 69 years old.

**Figure 4 genes-10-00735-f004:**
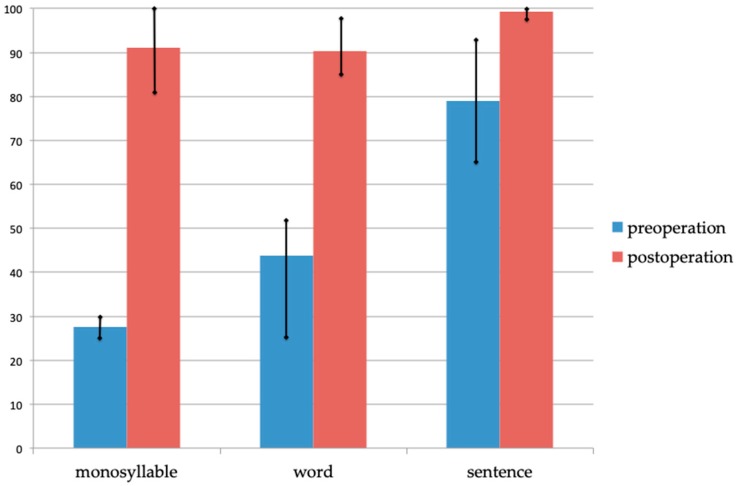
Mean scores of the Japanese monosyllable, word and sentence tests pre-cochlear implantation (CI) and six months post-CI. The scores pre-CI are with hearing aids. *n* = 2 pre-CI (*n* = 3 in word perception tests pre-CI) and *n* = 4 post-CI.

**Table 1 genes-10-00735-t001:** All possibly pathogenic *LOXHD1* gene variants identified in this study (NM_144612).

			Allele Frequency Information		ANNOVAR dbNSFP ver 3.5						
Base Change	AA Change	Pathogenecity	ExAC03	gnomAD	SIFT	PP2 HVAR	LRT	MutTaster	MutAssessor	CADD	Reference
c.246 − 1G > A		Pathogenic (PVS1,PM2,PM3,PP1,PP3)	-	-	-	-	-	D (0.81)	-	26.7	This Study
c.1270 + 4A > C		Likely Pathogenic (PVS1, PM2)	-	-	-	-	-	-	-	-	This Study
c.1828G > A	p.E610K	Uncertain Significance (PM3, PP3, BS1)	4.602e−05	5.3e−05	D (0.614)	D (0.85)	D (0.629)	D (0.548)	M (0.743)	25	[14] ＊
c.2726C > T	p.T909M	Uncertain Significance (PM2, BP4)	-	6.785e−06	D (0.443)	B (0.214)	N (0.383)	D (0.81)	L (0.263)	10.25	This Study
c.3076G > T	p.V1026F	Uncertain Significance (PM2, PM3, PP3, PP5)	-	-	D (0.721)	D (0.754)	D (0.537)	D (0.81)	H (0.932)	15.91	[15]
c.3281A > G	p.D1094G	Uncertain significance (PP3, PM7, PP3, PP5, BS1)	0	5.883e−05	D (0.555)	D (0.916)	D (0.843)	D (0.81)	H (0.973)	15.22	[16] ＊
c.3578C > T	p.A1193V	Uncertain Significance (PM2, PP3)	-	-	D (0.682)	D (0.818)	D (0.523)	D (0.81)	H (0.970)	33	This Study
c.3857G > T	p.G1286V	Likely Pathogenic (PM2, PM3, PP1, PP3)	-	-	D (0.784)	D (0.971)	D (0.743)	D (0.81)	H (0.967)	26.1	This Study
c.4212 + 1G > A		Pathogenic (PVS1, PM2, PM7, PP3, PP5)	-	2.65e−05	-	-	-	D (0.81)	-	24.2	[17]
c.4213 − 1G > A		Pathogenic (PVS1, PM2, PM3)	-	6.6e−06	-	-	-	D (0.81)	-	26	This Study
c.4375 + 1G > T		Likely Pathogenic (PVS1, PM2, PM3, PP5)	-	-	-	-	-	D (0.81)	-	26.6	[16]
c.4480C > T	p.E1494X	Pathogenic (PVS1, PM2, PM3, PP1, PP3, PP5)	0.0006	0.0006	-	-	D (0.843)	A (0.81)	-	38	[18,19,20,21]
c.4734C > G	p.Y1578X	Pathogenic (PVS1, PM2, PM3)	-	-	-	-	D (0.559)	A (0.81)	-	35	This Study
c.5086 − 3C > A		Pathogenic (PVS1, PM2, PP3)	-	6.623e−26	-	-	-	-	-		This Study
c.5608C > T	p.R1870W	Uncertain Significance (PM2, PP3)	4.499e−05	3.939e−05	D (0.784)	D (0.916)	-	D (0.48)	M (0.924)	34	This Study
c.5734G > A	p.D1912N	Uncertain Significance (PM2, PM3, PP3,)	-	-	D (0.912)	D (0.916)	-	D (0.588)	M (0.632)	28.7	This Study
c.5869G > T	p.E1957X	Pathogenic (PVS1, PM2, PM3, PP1, PP5)	-	-	-	-	-	A (0.81)		54	[21]
c.5885C > T	p.1962M	Uncertain Significance (PM2, PP1, PP3, PP5)	4.44e−05	6.544e−06	D (0.784)	D (0.916)	-	D (0.52)	L (0.44)	34	[22]
c.5933G > A	p.G1978D	Likely Pathogenic (PM2, PM3, PP1, PP3)	4.531e−05	6.565e−06	D (0.614)	D (0.679)	-	D (0.588)	L (0.52)	34	This Study
c.6037G > A	p.G2013R	Uncertain Significance (PM2, PM3, PP3)	-	-	D (0.682)	D (0.971)	-	D (0.81)	L (0.52)	33	This Study
c.6168delC	p.C2057Vfs * 42	Pathogenic (PVS1, PM2, PM3)	-	-	-	-	-	-	-	-	This Study

AA: amino acid; PVS: evidence of pathogenicity—Very strong; PM: evidence of pathogenicity—moderate; PP: evidence of pathogenicity—Supporting; BP: evidence of benign—Supporting; BS: evidence of benign—Strong. * This asterisk indicates the numbers of frameshifts. ＊ The MAF in the Japanese population were relatively higher than that of other populations. The pathogenicity of these variants is controversial.

**Table 2 genes-10-00735-t002:** Clinical findings of individuals with *LOXHD1* variants identified in this study (NM_144612).

Family No.	Base Change	Amino Acid Change	Onset	Sex	Tinnitus	Verigo	Vestibular Examination (R/L)	Progression	Severity	Configuration
1	c.[4212 + 1G > A]; [3857G > T]	p.[?]; [G1286V]	0	m	-	-	Normal/Normal	+	Mild	Flat
2	c.[4212 + 1G > A]; [4212 + 1G > A]	p.[?]; [?]	0	m	-	-	NA	-	Moderate	Flat
3	c.[4212 + 1G > A]; [4212 + 1G > A]	p.[?]; [?]	0	f	-	-	NA	+	Moderate	Precipitous
4	c.[4212 + 1G > A]; [4212 + 1G > A]	p.[?]; [?]	0	f	-	-	NA	-	Moderate	Flat
5	c.[4212 + 1G > A]; [4212 + 1G > A]	p.[?]; [?]	0	f	-	-	NA	-	Severe	Flat
6	c.[5885C > T]; [246 − 1G > A]	p.[1962M]; [?]	0	f	NA	NA	Normal/Normal	+	Profound	Flat
7 III-1	c.[4212 + 1G > A]; [4212 + 1G > A]	p.[?]; [?]	0	m	NA	-	NA	-	Moderate	Flat
III-2	c.[4212 + 1G > A]; [4212 + 1G > A]	p.[?]; [?]	0	f	NA	-	NA	-	Moderate	Flat
8 III-1	c.[5869G > T]; [4480C > T]	p.[E1957X]; [E1494X]	0	f	-	-	NA	+	Severe	Flat
III-2	c.[5869G > T]; [4480C > T]	p.[E1957X]; [E1494X]	7	m	-	-	NA	+	Moderate	Flat
9	c.[4213 − 1G > A]; [4212 + 1G > A]	p.[?]; [?]	5	m	-	-	NA	-	Mild	Sloping
10	c.[4375 + 1G > T]; [3076G > T]	p.[?]; [V1026F]	7	f	-	-	NA	-	Profound	Sloping
11	c.[4212 + 1G > A]; [3281A > G]	p.[?]: [D1094G]	0	f	-	-	NA	+	Severe	Sloping
12 III-1	c.[4212 + 1G > A]; [246-1G>A]	p.[?]; [?]	2	f	-	-	NA	+	Profound	Flat
III-2	c.[4212 + 1G>A]; [ 246 − 1G > A]	p.[?]; [?]	0	m	-	-	NA	+	Profound	Flat
13	c.[6168delC]; [4212 + 1G > A]	p.[C2057Vfs * 42]; [?]	3	m	+	-	Normal/Normal	+	Severe	Precipitous
14	c.[4734C > G]; [4212 + 1G > A]	p.[Y1578X]; [?]	0	m	-	-	NA	+	Profound	Sloping
15	c.[4212 + 1G > A]; [4212 + 1G > A]	p.[?]; [?]	4	f	NA	NA	NA	-	NA	NA
16	c.[5734G > A]; [3281A > G]	p.[D1912N]; [D1094G]	30	f	+	-	NA	+	Severe	Sloping
17	c.[3578C > T]; [3578C > T]	p.[A1193V]; [A1193V]	0	f	+	-	NA	-	Moderate	Precipitous
18	c.[6037G > A]; [4212 + 1G > A]	p.[G2013R]; [?]	5	f	+	-	Normal/Normal	+	Profound	Sloping
19	c.[5933G > A]; [3281A > G]	p.[G1978D]; [D1094G]	32	f	+	-	NA	+	Profound	Precipitous
20	c.[4212 + 1G > A]; [4212 + 1G > A]	p.[?]; [?]	0	f	+	-	NA	-	Profound	Flat
21	c.[5608C > T]; [1270 + 4A > C]	p.[R1870W]; [?]	36	m	NA	NA	NA	+	Mild	Precipitous
22	c.[5086 − 3C > A]; [2726C > T]	p.[?]; [T909M]	30	f	+	-	Normal/Normal	+	Profound	Precipitous
23	c.[4212 + 1G > A]; [3281A > G]	p.[?]: [D1094G]	4	f	NA	NA	Normal/Normal	+	Severe	Sloping
24	c.[1828G > A]; [1828G > A]	p.[E610K]; [E610K]	2	f	NA	NA	NA	+	Profound	Flat
25	c.[4212 + 1G > A]; [4212 + 1G > A]	p.[?]; [?]	5	m	-	-	NA	-	Profound	Flat

NA: not available, -: No, +: Yes. * This asterisk indicates the numbers of frameshifts.

**Table 3 genes-10-00735-t003:** All previously reported *LOXHD1* gene variations and their clinical features (NM_144612).

Nucleotide Change	Amino Acid Change	HL Onset	Severity of HL	Progression	Population	Reference
c.[71delT]; [71delT]	p.[L24Rfs * 74]; [L24Rfs * 74]	Congenital or Prelingual	Severe or Profound	NA	Turkish	[23]
c.[442A > T]; [4217C > T]	p.[K148X]; [A1406V]	NA	NA	NA	American	[24]
c.[486_487delCTinsGG]; [486_487delCTinsGG]	p.[?]; [?]	NA	NA	NA	Saudi Arabian	[25]
c.[894T > G]; [6353G > A]	p.[Y298X]; [G2118E]	Congenital	Mild-moderate	NA	American	[19]
c.[1588C > T]; [1588C > T]	p.[E530X]; [E530X]	Childhood	Severe–profound	Progressive	Qatar	[26]
c.[1603C > T]; [1938G > A]	p.[R535X]; [K646K]	Childhood	Mild–moderate	NA	American	[19]
c.[1618dupA]; [1730T > G]	p.[T540Afs * 24]; [L635P]	Congenital	Severe	Stable–Progressive	Dutch	[14,22]
c.[1751C > T]; [5815G > A]	p.[T584]; [D1939N]	35–40 y.o.	Severe	Progressive	Chinese	[27]
c.[1730T > G]; [5869G > A]; [5944C > T] ＊	p.[L577R]; [E1957K]; [R1982X]	Congenital	Severe–profound	NA	American	[19]
c.[1828G > T]; [2641G > A]	p.[E610X]; [G881R]	2–4 y.o.	Mild	Stable	Dutch	[14]
c.[1843C > T]; [3281A > G]	p.[R615W]; [D1094G]	NA	NA	NA	Chinese	[16]
c.[1904T > C]; [4678T > C]	p.[L635P]; [C1560R]	2–3 y.o.	Mild	Stable–Progressive	Dutch	[14]
c.[1938G > A]; [4936C > T]	p.[K646K]; [C1560R]	Childhood	Mild–moderate	NA	American	[19]
c.[2008C > T]; [2008C > T]	p.[R670X]; [R670X]	Childhood	Moderate–profound	Progressive	Iranian	[5]
c.[2696G > C]; [3596T > C]	p.[R899P]; [L1199P]	NA	NA	NA	American	[19]
c.[2696G > C]; [3834G > C]	p.[R899P]; [W1278C]	5 y.o.	Moderate	Stable	Dutch	[14]
c.[2696G > C]; [5934C > T]	p.[R899P]; [T1978G]	Congenital	Mild	NA	Dutch	[14]
c.[2825_2827delAGA]; [4217C > A]	p.[?]; [A1406E]	Childhood	Mild–Moderate	NA	American	[19]
c.[2863G > T]; [2863G > T]	p.[E955X]; [E955X]	NA	NA	NA	Turkey	[18]
c.[3061C > T]; [5885C > T]	p.[R1021X]; [T1962M]	1–10 y.o.	Severe	Progressive	Netherlands	[14,22]
c.[3061 + 1G > A]; [6353G > A]	p.[?]; [G2118E]	Congenital	Moderate	NA	Netherlands	[14,22]
c.[3071A > G]; [3071A > G]	p.[Y1024C]; [Y1024C]	Early-onset	Severe–profound	NA	Italy	[28]
c.[3076G > T]; [4375 + 1G > T]	p.[V1026F]; [?]	3 y.o.	Profound	Non-progressive	Japanese	[15]
c.[3169C > T]; [6353G > A]	p.[V1026F]; [G2118E]	Congenital	Severe	Stable	Dutch	[14]
c.[3371G >A]; [3979T > A]	p.[A1124H]; [P1327I]	NA	NA	NA	Cameroon	[29]
c.[3571A > G]; [3571A > G]	p.[T1191A]; [T1191A]	Congenital	Severe-profound	NA	Spanish	[30]
c.[3748 + 1G > C]; [6353G > A]	p[?]; [G2118E]	Congenital	Moderate–severe	Stable–Progressive	Dutch	[14]
c.[4099G > T]; [6162_6164delCCT]	p.[E1367X]; [F2055Nfs * 157]	Congenital	Severe–profound	NA	American	[19]
c.[4212 + 1G > A]; [5674G > T]	p.[?]; [V1892F]	Congenital–7 y.o.	Mild–Severe	Progressive	Japanese	[17]
c.[4480C > T]; [4480C > T]	p.R1494X[R1494X]	NA	NA	NA	Turkey	[18]
c.[4480C > T; 4526G > A]; [4480C > T; 4526G > A]	p.[R1494X;G1509E]; [R1494X;G1509E]	Congenital	Mild–moderate	NA	American	[19]
c.[4480C > T]; [4526G > A]	p.[R1494X]; [G1509E]	40 y.o.	Severe–profound	Progressive	American	[20]
c.[4480C > T]; [5869G > T]	p.[R1494X]; [E1957X]	NA	Moderate–severe	Non-progressive	Japanese	[21]
c.[4480C > T; 4526G > A]; [6598delG]	p.[R1494X; G1509E]; [D2200Mfs * 21]	Childhood	Severe-profound	NA	American	[19]
c.[4623C >G]; [5545G > A]	p.[Y1541X]; [G1849R]	2 y.o.	Severe	NA	Czech	[31]
c.[4714C >T]; [4714C > T]	p.[R1572X]; [R1572X]	Prelingual	Severe–profound	Non-progressive	Ashkenazi Jews	[32]
c.[5894dupG]; [5894dupG]	c.[?]; [?]	Prelingual	Profound	Na	Arab	[33]
c.[5948C > T]; [5948C > T]	p.[S1983F]; [S1983F]	Congenital	Severe–profound	Non-progressive	Chinese	[34]

HL: hearing loss; NA: not available. ＊ Phasing of the three variants were unknown. * This asterisk indicates the numbers of frameshifts.

**Table 4 genes-10-00735-t004:** Haplotype analysis of *LOXHD1* recurrent variation, c.4212 + 1G > A.

Distance from the c.4212 + 1G > A Variation (bp)	Allele Frequency (HapMap-JPT)	Marker	No. 2	No. 3	No. 4	No. 5	No. 7	No. 25
980,731	C 0.30 T 0.70	rs868409	C/T	C/T	C/C	C/T	C/T	C/C
926,737	A 0.38 G 0.62	rs4890557	A/G	A/G	G/G	A/G	A/A	G/G
877,703	A 0.27 G 0.73	rs17732049	A/G	G/G	G/G	G/G	A/G	G/G
862,524	T 0.26 C 0.74	rs3786397	C/C	T/C	C/C	T/C	C/C	C/C
860,937	T 0.35 C 0.65	rs1552329	T/C	C/C	C/C	T/C	C/C	C/C
811,321	G 0.37 A 0.63	rs3760578	G/G	A/A	A/A	A/A	A/A	A/A
668,955	T 0.26 C 0.74	rs12456289	C/C	C/C	C/C	C/C	C/C	C/C
626,721	C 0.26 T 0.74	rs8097963	T/T	T/T	T/T	T/T	T/T	T/T
570,604	A 0.28 T 0.72	rs8087546	T/T	T/T	T/T	T/T	T/T	T/T
524,148	A 0.32 G 0.68	rs673123	G/G	A/G	G/G	G/G	G/G	G/G
332,094	C 0.47 T 0.53	rs9956574	C/T	C/T	T/T	C/T	T/T	T/T
233,342	C 0.38 T 0.62	rs4890637	T/T	T/T	T/T	C/T	T/T	T/T
134,221	T 0.32 C 0.68	rs16978558	C/C	C/C	C/C	C/C	C/C	C/C
122,941	C 0.27 A 0.73	rs3911131	C/A	A/A	A/A	A/A	A/A	A/A
24,152	G 0.27 T 0.73	rs8084298	T/T	T/T	T/T	T/T	T/T	T/T
15,561	A 0.29 G 0.71	rs426303	G/G	G/G	G/G	G/G	G/G	G/G
0		c.4212 + 1 G > A	-	-	-	-	-	-
108,292	A 0.23 C 0.77	rs16939868	C/C	C/C	C/C	C/C	C/C	C/C
206,155	C 0.27 G 0.73	rs4121822	C/C	C/C	C/G	C/G	C/G	C/C
220,199	G 0.27 A 0.73	rs4449041	A/A	A/A	G/G	A/A	G/G	A/A
282,037	C 0.28 T 0.72	rs513775	T/T	T/T	T/T	C/T	T/T	T/T
290,228	A 0.32 T 0.68	rs578451	T/T	T/T	A/T	A/T	T/T	T/T
296,452	A 0.26 G 0.74	rs2576050	G/G	G/G	G/G	A/G	G/G	G/G
444,615	A 0.28 G 0.72	rs2576040	G/G	G/G	G/G	A/G	G/G	G/G
538,731	G 0.26 A 0.74	rs16949034	A/A	A/A	G/A	A/A	G/A	A/A
615,845	A 0.27 G 0.73	rs1434529	G/G	G/G	G/G	G/G	G/G	G/G
631,424	C 0.24 G 0.76	rs1398218	G/G	G/G	G/G	G/G	G/G	G/G
733,575	C 0.35 T 0.65	rs1434506	C/T	C/C	C/T	C/T	T/T	C/C
805,030	C 0.33 T 0.67	rs1893784	C/T	T/T	C/T	T/T	C/T	T/T
822,756	C 0.39 T 0.61	rs4986222	C/C	C/C	C/T	C/T	C/T	C/C
973,908	C 0.36 T 0.64	rs1108062	T/T	T/T	T/T	T/T	C/T	T/T
1,042,874	G 0.33 A 0.67	rs3813071	A/A	A/A	G/A	A/A	A/A	A/A
1,100,150	T 0.30 G 0.70	rs12969708	G/G	G/G	T/G	G/G	G/G	G/G

Pink squares indicate heterozygotes and blues indicate homozygotes of minor allele.

## Data Availability

The sequencing data are available in the DDBJ databank of Japan (Accession number: JGAS00000000192).

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
