# Peer review of "Mutational Spectrum and Clinical Features of Patients with LOXHD1 Variants Identified in an 8074 Hearing Loss Patient Cohort"

_genes, 2019, doi:10.3390/genes10100735_

Round 1

Reviewer 1 Report

Thank you for the opportunity to review the manuscript by Maekawa et al entitled “Mutations Spectrums and Clinical Features of Patients with LOXHD1 Variants Identified in a 8,074 Hearing Loss Patient Cohort”. This is a nice analysis and provides detailed insights into the genotype and phenotype of this important gene that contributes to the pathogenesis of human hearing loss. 

While this paper is very good, evaluates a large cohort, and nicely drafted there are some important issues that detract from the research conclusions:

1) Next generation sequencing data: While the authors only presented LOXHD1 variants found in their deaf cohort, they did not present the full dataset of other pathogenic, likely pathogenic, or unknown variants in other hearing loss genes that were also found in these patients. This data is important, especially as some of the pedigrees lack evidence of consanguinity and look to be dominant or X-linked, yet the authors are evoking a recessive mechanism of disease. This is especially important  for most of the families, segregation analysis as not performed (see comment below). For those of us who have sequenced hundreds of deaf individuals using similarly sized panels, there are frequently variants in more than one gene per individual.

2) Next generation methodology: The authors should provide the gene list for their 68 gene hearing loss panel. They should also state the work arounds that were performed for the large GJB6 deletion and the STRC deletion if they were performed. They should also state if copy number variant analysis was performed for other genes. 

3) Segregation analysis: For the pedigrees in figure 1, segregation analysis was not performed for many families. This is an issue, especially since the c.4212+1G>A variant may have arisen on multiple haplotypes and suggesting that it could occur de novo. It would be very helpful if we know that each variant in each individual occurred on a separate chromosome. It is not longer safe to assume that if an individual has two variants in a gene, that they exist in trans. If parents are not available for the many probands, long read sequencing should be considered to set phase.

4) Grammar and usage: While this is a exceedingly well drafted manuscript, there are still a few English grammar and usage issues. For example in the title, it should read “Spectrum” rather than “Spectrums”.

Author Response

Thank you for the opportunity to review the manuscript by Maekawa et al entitled “Mutations Spectrums and Clinical Features of Patients with LOXHD1 Variants Identified in a 8,074 Hearing Loss Patient Cohort”. This is a nice analysis and provides detailed insights into the genotype and phenotype of this important gene that contributes to the pathogenesis of human hearing loss. While this paper is very good, evaluates a large cohort, and nicely drafted there are some important issues that detract from the research conclusions:

1) Next generation sequencing data: While the authors only presented LOXHD1 variants found in their deaf cohort, they did not present the full dataset of other pathogenic, likely pathogenic, or unknown variants in other hearing loss genes that were also found in these patients. This data is important, especially as some of the pedigrees lack evidence of consanguinity and look to be dominant or X-linked, yet the authors are evoking a recessive mechanism of disease. This is especially important for most of the families, segregation analysis as not performed (see comment below). For those of us who have sequenced hundreds of deaf individuals using similarly sized panels, there are frequently variants in more than one gene per individual.

>Thank you for your comment. We understand the importance to indicate other gene mutations identified in these families, however, in this study, we focused on showing the mutation spectrum and clinical characteristics of LOXHD1gene associated hearing loss. In addition, most of the papers published in this field did not include such full mutation list. Instead of a full mutation list, we reposited fastq files for each patient in a public database so readers can access the appropriateness of our analysis.

As noted in the paper, the MPS did not detect any other possible variants to be the cause of HL for all the patients. We re-reviewed all the remaining pathogenic, likely pathogenic, and uncertain significant variants identified in other genes after the filtering process (written in the Methods section) again, however, we couldn’t detect any other biallelic candidate variants in other recessive genes. Specifically, families #17, #21 and #22 looks to be possibly of dominant or X-linked heridity, only the LOXHD1mutations remained after filtering, thus we included these three cases. We mentioned this clearly in the Result section.

2) Next generation methodology: The authors should provide the gene list for their 68 gene hearing loss panel. They should also state the work arounds that were performed for the large GJB6 deletion and the STRC deletion if they were performed. They should also state if copy number variant analysis was performed for other genes. 

>Thank you for your valuable comment. We have made a list of the 68 genes and attached the table as a supplementary material. We have also modified the paper so as to clarify about the copy number variant analysis.

3) Segregation analysis: For the pedigrees in figure 1, segregation analysis was not performed for many families. This is an issue, especially since the c.4212+1G>A variant may have arisen on multiple haplotypes and suggesting that it could occur de novo. It would be very helpful if we know that each variant in each individual occurred on a separate chromosome. It is not longer safe to assume that if an individual has two variants in a gene, that they exist in trans. If parents are not available for the many probands, long read sequencing should be considered to set phase.

>We were not able to obtain the peripheral blood samples of the families of those that the segregation analysis was not performed. Of course, by using the long read sequencer, it is possible to reveal the cis or trans location of these variants, however, it was impossible to perform in this short revision period. We have modified the paper to comment on this fact and about the limitations.

4) Grammar and usage: While this is a exceedingly well drafted manuscript, there are still a few English grammar and usage issues. For example in the title, it should read “Spectrum” rather than “Spectrums”.

>We have reviewed the paper again, modified it as suggested, and checked for other grammatical issues.

Reviewer 2 Report

The article presents a description of the phenotypes of variants in the LOXHD1 gene in a sample of Japanese people with hearing loss. It is generally well written and presented, for which I thank the authors. Except for one place, which I mention below, the English is good, or at least comprehensible. I do not have many comments to make, and none are greatly significant.

L57: The phrase “, , , and over 60% of prelingual HL cases account for genetic factors” is hard to follow and is not supported by the citation that follows it as far as I can tell. Please can the authors reword this and provide a more suitable reference.

L123: Clinical evaluations must have used different techniques across ages – evoked potentials, VRA, etc. Please could the authors indicate this. If there are any cases where earlier evoked potentials can be compared to later behavioural measures, perhaps the authors could comment on this?

Figure 1: This is missing a key to m/f, alive/dead/, with/without. I wonder if it would make the figure more meaningful to order the cases by age at testing and degree of hearing loss. Perhaps the authors could try this and/or other approaches to making the figure more systematic?

Figure 3: It looks as though the age split is arbitrary. One line in part a is similar to the data in part b, and measured across more frequencies, suggesting that it is an older child. I presume that the other lines were tested using AEPs or other forms of behavioural audiometry. Perhaps that could be indicated, and maybe try presenting the data in other ways: for example using the graded darkness approach for age that u=you used in an earlier figure?

Author Response

The article presents a description of the phenotypes of variants in the LOXHD1 gene in a sample of Japanese people with hearing loss. It is generally well written and presented, for which I thank the authors. Except for one place, which I mention below, the English is good, or at least comprehensible. I do not have many comments to make, and none are greatly significant.

L57: The phrase “, , , and over 60% of prelingual HL cases account for genetic factors” is hard to follow and is not supported by the citation that follows it as far as I can tell. Please can the authors reword this and provide a more suitable reference.

>Thank you for the valuable comment. We have checked the citation, and modified it as suggested.

L123: Clinical evaluations must have used different techniques across ages – evoked potentials, VRA, etc. Please could the authors indicate this. If there are any cases where earlier evoked potentials can be compared to later behavioural measures, perhaps the authors could comment on this?

>Evaluation of HL by pure-tone audiometry was performed on patients over 5 years old, and auditory steady state response (ASSR), conditioned orientation response audiometry (COR: one of the behavioral audiometry), or play audiometry was performed on patients under 4 years old.

Figure 1: This is missing a key to m/f, alive/dead/, with/without. I wonder if it would make the figure more meaningful to order the cases by age at testing and degree of hearing loss. Perhaps the authors could try this and/or other approaches to making the figure more systematic?

>We have modified the paper as suggested.

Figure 3: It looks as though the age split is arbitrary. One line in part a is similar to the data in part b, and measured across more frequencies, suggesting that it is an older child. I presume that the other lines were tested using AEPs or other forms of behavioural audiometry. Perhaps that could be indicated, and maybe try presenting the data in other ways: for example using the graded darkness approach for age that u=you used in an earlier figure?

>Thank you for your comment. Most of the data of figure 3a was of behavioural audiometry (conditioned orientation response audiometry) and one was of pure tone audiometry. We remade the figure as suggested and added some explanations.

Round 2

Reviewer 1 Report

The authors were responsive to the reviews.